



# Photomineralization mechanism changes the ability of dissolved organic matter to activate cloud droplets and to nucleate ice crystals

Nadine Borduas-Dedekind[1,2], Rachele Ossola[1], Robert O. David[2], Lin S. Boynton[1], Vera Weichlinger[2], Zamin A. Kanji[2], and Kristopher McNeill[1]

[1]Institute for Biogeochemistry and Pollutant Dynamics, ETH Zurich, Zurich, 8092, Switzerland
[2]Institute for Atmospheric and Climate Sciences, ETH Zurich, Zurich, 8092, Switzerland

*Correspondence to*: Nadine Borduas-Dedekind (nadine.borduas@usys.ethz.ch); @nadineborduas and *to* Zamin A. Kanji (zamin.kanji@env.ethz.ch)

**Abstract**

An organic aerosol particle has a lifetime of approximately one week in the atmosphere during which it will be exposed to sunlight. Yet, the effect of photochemistry on the propensity of organic matter to participate in the initial cloud-forming steps is difficult to predict. In this study, we quantify on a molecular scale the effect of photochemical exposure of naturally occurring dissolved organic matter (DOM) and of a fulvic acid standard on its ability to form mixed-phase clouds, by acting as cloud condensation nuclei (CCN) and by acting as ice nucleating particles (INPs). We find that photochemical processing, equivalent to 4.6 days in the atmosphere, of DOM increases its ability to form cloud droplets by up to a factor of 2.5 but decreases its ability to form ice crystals at a loss rate of $-0.04\ °C_{T50}\ h^{-1}$ of sunlight at ground level. In other words, the ice nucleation activity of photooxidized DOM can require up to 4 degrees colder temperatures for 50% of the droplets to activate as ice crystals under immersion freezing conditions. This temperature change could impact the ratio of ice to water droplets within a mixed-phase cloud by delaying the onset of glaciation and by increasing the supercooled liquid fraction of the cloud, thereby modifying the radiative properties and the lifetime of the cloud. Concurrently, a photomineralization mechanism was quantified by monitoring the loss of organic carbon and the simultaneous production of organic acids, such as formic, acetic, oxalic and pyruvic acids, CO and $CO_2$. This mechanism explains and predicts the observed increase in CCN and decrease in INP efficiencies. Indeed, we show that photochemical processing can be a dominant atmospheric aging process, impacting CCN and INP efficiencies and concentrations. Photomineralization can thus alter the aerosol-cloud radiative effects of organic matter by modifying the supercooled liquid water-to-ice crystal ratio in mixed-phase clouds with implications for cloud lifetime, precipitation patterns and the hydrological cycle.

**Significance statement**

During atmospheric transport, dissolved organic matter (DOM) within aqueous aerosols undergoes photochemistry. We find that photochemical processing of DOM increases its ability to form cloud droplets but decreases its ability to form ice crystals



over a simulated 4.6 days in the atmosphere. A photomineralization mechanism involving the loss of organic carbon and the production of organic acids, CO and $CO_2$ explains the observed changes and affects the liquid water to ice ratio in clouds.

## 1. Introduction

Aerosol-cloud interactions play a key role in the earth's energy budget, yet contribute to the largest uncertainty in radiative forcing in climate model estimates (Boucher et al., 2013). Mixed-phase clouds are particularly interesting because of their predominant role in global precipitation and their unstable microphysics due to the co-existence of liquid water and ice. The presence of liquid water in mixed-phase clouds is due to the ability of an aerosol particle to activate into a cloud droplet in the presence of high relative humidity, as predicted by κ-Köhler theory (Farmer et al., 2015; Petters and Kreidenweis, 2007). First the size and second the chemical composition of aerosol particles are the key factors controlling their ability to act as cloud condensation nuclei (CCN) (Dusek et al., 2006; Köhler, 1936). The initial formation of ice crystals in mixed-phase clouds is due to the ability of an ice nucleating particle (INP) to induce freezing in a supercooled liquid droplet via heterogeneous ice nucleation predominantly through the immersion freezing mechanism (Knopf et al., 2018; Vali, 2014). Furthermore, ice crystals grow at the expense of water droplets in mixed-phase clouds because the saturation water vapor pressure is lower over ice than over water (Korolev, 2007). In this study, we chose to combine cloud condensation and ice nucleation experiments on the same particle type to elucidate effects on mixed-phase clouds with implications for the subtle balance existing between supercooled water droplets and ice crystals.

This study focuses on the impact of atmospheric photochemical processing during the one-week lifetime equivalent of an organic aerosol particle. Indeed, in the time interval between where organic aerosols are emitted and/or formed and where they act as CCN and INP, they will undoubtedly be exposed to sunlight and thus undergo atmospheric processing through photochemistry (George et al., 2015; Laskin et al., 2015). This atmospheric aging process modifies the physical and chemical properties of organic aerosols and subsequently affects their cloud-forming ability. In fact, atmospheric photochemical processing of organic aerosols has been shown to increase CCN ability (Slade et al., 2017; Wong et al., 2011). However, the effect of photochemical processing of organic matter on ice nucleation is unknown, although recent work on pollen in deposition freezing mode suggests a decrease in ice activity (Gute and Abbatt, 2018). Our study links a photochemical mechanism with CCN and with INP efficiencies applicable for an atmospheric organic aerosol.

Aquatic dissolved organic matter (DOM) is known to be highly photoreactive by direct and indirect photochemical processes (McNeill and Canonica, 2016; Rosario-Ortiz and Canonica, 2016; Sharpless et al., 2014; Wenk et al., 2013). DOM therefore allows us to probe the impact of photochemistry on the cloud forming properties of organic matter, while taking advantage of its known photochemical processes in aquatic environments. Furthermore, both DOM and organic aerosols possess absorbing





molecules in the visible spectrum, often termed brown carbon (Laskin et al., 2015). Both materials are also naturally occurring and contain complex organic molecules, with DOM having more chemical function group diversity (Graber and Rudich, 2006; Kristensen et al., 2015).

## 2. Materials and methods

### 2.1. Sample description and storage

The DOM samples were collected from the Great Dismal Swamp, in Virginia, USA, and from the Suwannee River, in Florida, USA. The Great Dismal Swamp was sampled in Suffolk (36.7˚S, 76.4˚W) in 2014 (Sun et al., 2014) and from Jericho Ditch (36.7˚S, 76.4˚W) in 2016 (Lin et al., 2017). Suwannee River water was collected in 2017 close to the collection site of the International Humic Standard Society (IHSS) (30.5˚N, 82.5˚W). The Dismal Swamp and the Suwannee River samples were filtered through a pre-cleaned 0.2 µm capsule filter (Polycap TC, Whatman) on site following collection and kept refrigerated until use. The samples were never frozen to avoid potential freeze-thaw cycle artefacts. Experimental solutions with concentrations of approximately 20 mg C L$^{-1}$ were prepared upon dilution of the field-collected waters in nanopure water. Suwannee River Fulvic Acid Standard II (2S101F) was purchased from the IHSS and dissolved in nanopure water with concentrations of approximately 20 mg C L$^{-1}$. The inorganic ions present in Dismal Swamp water have been previously reported (Johannesson et al., 2004) and anionic ion chromatography confirmed similar concentrations. Indeed, DS2014 had concentrations of F$^-$, Cl$^-$, SO$_4^{2-}$, NO$_3^-$ for a total of 30 µM, which we can approximate to 60 µM of ions assuming charge balance. These ion concentrations suggest that the ratio of organic to inorganic ions is typically less than 10% by molar concentration.

In addition, the DOM sampled for this study were bulk water-collected and therefore represent an alternative sampling procedure to access organic carbon compared to impacting particular matter on filters. Indeed, DOM was not extracted from a filter nor from an impinger, but rather directly collected in the field. Ultimately, the photochemical mechanisms identified in this study are expected to be similar between DOM and organic aerosols, thereby bridging the fields of aquatic and atmospheric chemistry of organic matter.

### 2.2. Photochemical setup

Photolysis experiments were conducted in a commercial photoreactor (Rayonet, Southern New England Ultraviolet Co) equipped with a motorized turn-table and 6 × 300 nm light bulbs (UVB – 3000 Å from Southern New England Ultraviolet Co.). These UVB bulbs have a relative light intensity as function of wavelength shown in Fig. S2. 9 mL of a 20 mg C L$^{-1}$ DOM or IHSS isolate solution was pipetted into cork-capped 10-mL borosilicate test tubes (Pyrex, 15 × 85 mm, disposable) and irradiated for up to 25 h. During this period, the temperature inside the photoreactor was held at 30-32˚C and the reactor was turned on prior to the beginning of the experiment to ensure a constant temperate exposure. At each time point, a test tube was



removed from the photoreactor, and the solution was directly used for either CCN (9-18 mL), INP (9-18 mL), TOC (7 mL), ion chromatography (1 mL) or conductivity (0.1 mL) measurements.

A slightly different experimental setup was used for the CO and $CO_2$ measurements. In order to avoid $CO_2$ partitioning to the

gas phase, the borosilicate test tubes were prepared headspace-free using rubber septa. At each time point, a test tube was withdrawn from the photoreactor and 6 mL of the experimental solution were transferred to a $N_2$-flushed 20 mL serum vial containing 100 µL HCl 1N. The serum vial was briefly shaken and stored no longer than a week in the fridge until headspace analysis on the GC-FID.

### 2.3. Photochemical control experiments

Control experiments were conducted to unambiguously attribute the change in CCN and INP to photochemistry. Importantly, the photomineralization mechanism could have led to the accumulation of carbonate ions in solution and thus increase the CCN ability of the material with UVB exposure time. As a control, inorganic carbon measurements by the Shimadzu TOC analyser showed no increasing IC fraction with irradiation. In addition, control experiments where the DOM solutions were purged with argon showed no difference in CCN activity, indicating no contribution to accumulating dissolved carbonate in

solution.

To address the issue of the potential impact of a 30 °C heat exposure on CCN and INP during the photochemical experiment, foil-covered test tubes were placed inside the photoreactor alongside non-covered test tubes. Indeed, the non-irradiated sample that would have experienced the same temperature elevation from room temperature to 30 °C, did not show any change in

hygroscopicity, INP activity, organic carbon nor production of photochemical adducts.

Control experiments involving Ar-purged solutions further emphasize that dissolved $O_2$ is necessary for photomineralization, consistent with previous reports (Schmitt-Kopplin et al., 1998). In an atmospheric context, $O_2$ will be readily available, to the aqueous aerosol phase and we expect in situ aqueous photomineralization of organic aerosols. Furthermore, as confirmed by

dark control experiments, exposure to light was necessary to drive the chemical, cloud droplet formation and ice nucleation changes observed.

### 2.4. Actinometry experiments for relating UVB exposure to sunlight

The light intensity inside the photoreactor was monitored with the chemical actinometer pyridine/*p*-nitroanisole (PNA) (Laszakovits et al., 2017). Briefly, a solution containing 20 µM of recrystallized PNA and 0.25 mM of pyridine in nanopure

water was irradiated for 5 h in the experimental conditions described above. Then, the PNA and pyridine were quantified via ultra-high-pressure liquid chromatography (Waters ACQUITY) equipped with a C18 column (Aquity, BEH130 C18, 1.7 µm;



2.1 × 150 mm) and a photodiode array detector. The analyses were performed using a 40:60 A:B eluent mixture of HPLC grade solvent (A = 90% (acetate buffer pH 6) + 10% acetonitrile; B = 100% acetonitrile), 5 µL injection volume and 0.2 mL/min flow rate. At these conditions, PNA was eluted at 4.2 min (detection at 310 nm) and pyridine at 2.7 min (detection at 250 nm).

The absolute spectral irradiance ($I_\lambda$) was obtained according to $I_\lambda = s \cdot I_{\lambda,m}$, where $I_{\lambda,m}$ (mE cm$^{-2}$ s$^{-1}$ nm$^{-1}$) is the spectral irradiance measured in the photoreactor with a calibrated Jaz spectrophotometer (Ocean Optics) using 2 × 300 nm bulbs (Fig. S2). $I_{\lambda,m}$ was also corrected for the absorption of the borosilicate glass occurring between 280-300 nm. The scaling factor $s$ was calculated according to Eq. (1):

$$s = \frac{k_{deg,\text{PNA}}[\text{PNA}]_0 l}{\Phi_{deg,\text{PNA}} \sum_\lambda I_{\lambda,m} f_{\lambda,\text{PNA}} \Delta\lambda} \tag{1}$$

where $k_{deg,\text{PNA}} = (0.37 \pm 0.02)$ h$^{-1}$ is the pseudo-first-order PNA degradation rate constant measured with 6 × 300 nm bulbs, $[\text{PNA}]_0 = (17.4 \pm 0.4)$ µM is the starting PNA concentration, $\Phi_{\text{PNA}} = 0.076 \pm 0.008$ is the direct photolysis quantum yield calculated according to (Laszakovits et al., 2017), $f_{\lambda,\text{PNA}}$ is the absorptivity of the starting PNA solution, $l$ is the path length ($l$

= 1 cm) and $\Delta\lambda = 1$ nm. $k_{deg,\text{PNA}}$, $[\text{PNA}]_0$ and $\Phi_{\text{PNA}}$ are the average values measured for different days. $f_{\lambda,\text{PNA}}$ represents the fraction of light absorbed by the PNA starting solution, and it was calculated according to $f_{\lambda,\text{PNA}} = 1 - 10^{-\varepsilon_{\lambda,\text{PNA}}[\text{PNA}]_0 l} = 1 - 10^{-A_{\lambda,\text{PNA}}}$, where $\varepsilon_{\lambda,\text{PNA}}$ is the molar extinction coefficient of PNA and $A_{\lambda,\text{PNA}}$ is the absorbance. This calculation provided an integrated irradiance (280 – 400 nm) of 64 ± 4 J s$^{-1}$ m$^{-2}$ for 6 x 300 nm bulbs. The photoreactor-to-sunlight conversion factor, that is, the irradiation time in the photoreactor equivalent to one hour in natural sunlight, was obtained as the ratio of

the rates of light absorption of a given DOM for the two different light sources ($R^{abs}_{\text{photoreactor}}$, $R^{abs}_{\text{solar}}$), according to Eq. 2.

$$conversion\ factor = \frac{R^{abs}_{\text{photoreactor}}}{R^{abs}_{\text{solar}}} \tag{2}$$

In order to get $R^{abs}_{\text{solar}}$, we simulated a solar spectrum (Global Horizontal Irradiance, 300 – 400 nm, integrated irradiance of 59.2 J s$^{-1}$ m$^{-2}$) using SMARTS 2.9.5 (NREL) for a generic point at middle latitudes (45˚) in summer (1$^{st}$ June, from 5 a.m. to 7 p.m.). Using the daily average value of 2.2 during daytime, 25 hours of irradiation in the photoreactor with 6 UVB bulbs

corresponds to 55 hours of sunlight irradiation, which are equivalent to 4.6 days in the environment (assuming 12 daily hours of light on a clear day). The actinometry experiments are independent of temperature, as they depend on photon flux.

### 2.5. Analytical chemistry instruments

#### 2.5.1. TOC analyzer

The total organic carbon, specifically the non-purgeable organic carbon (NPOC), and the total inorganic carbon (IC,

carbonates) were quantified using a total organic carbon (TOC) analyser (Shimadzu, model TOC-L CSH). The NPOC method



used 50 μL injections for a minimum of triplicate measurements and reports concentrations in mg C L$^{-1}$ with standard deviations from the average. Method purge time and gas flow were 1.5 mins and 80 mL, respectively. NPOC calibrations were done with recrystallized potassium phthalate. The IC calibration was done with potassium carbonate ($K_2CO_3$) solutions and the IC method also used 50 μL injections for a minimum of triplicate measurements. To confidently attribute the decreasing

TOC value observed during irradiation to the formation of gaseous products (CO and $CO_2$), we ran solutions containing formaldehyde, formic acid, acetone and urea with the NPOC method and we confirmed that those compounds were not removed from the solutions during the purging time within the TOC instrument.

### 2.5.2. Conductivity measurements

Conductivity measurements were made using a portable Horiba Scientific LAQUAtwin model B-771 conductivity kit. The

conductivity probe was calibrated with the manufacturer's standard solution at 1.41 mS/cm.

### 2.5.3. GC-FID measurements

CO and $CO_2$ headspace concentrations were quantified by a gas chromatograph (GC) equipped with a flame ionization detector (FID) and a methanizer (Model 8610, SRI instruments, Menlo Park CA). The methanizer ensures high sensitivity following separation over a 9 ft Hayesep D column with $N_2$ carrier gas. Column and detector temperatures were 40 °C and 300 °C,

respectively. Under these conditions, CO and $CO_2$ eluted at 1.82 min and 4.27 min, respectively. From the headspace concentration ($p_{A,hs}$, where A is CO or $CO_2$), the aqueous phase concentration in the photolysis test tube ($[A]_{aq}^0$) was calculated with Eq. (3).

$$[A]_{aq}^0 = \frac{n_{A,aq} + n_{A,hs}}{V_{aq}} \qquad (3)$$

where $V_{aq}$ is the volume of liquid in the 20 mL serum vial, and $n_{A,aq}$ and $n_{A,hs}$ are the moles of $CO_2$ in the aqueous phase and

in the headspace, respectively. $n_{A,aq}$ was calculated from the measured headspace vapour pressure ($p_{A,hs}$) via Henry's law, while $n_{A,hs}$ was also obtained from $p_{A,hs}$ using the ideal gas law and Eq. (4).

$$n_{A,aq} = \frac{p_{A,hs}}{K_A} \cdot V_{aq} \qquad (4)$$

$K_A$ is the Henry's constant corrected for the temperature $T$ at the time of the measurement (Fry et al., 1995), $R$ is the gas constant and $V_{hs,corr}$ is the headspace volume corrected for the vial overpressure ($V_{hs,corr} = V_{hs} \cdot \frac{p_{tot,hs}}{p_{atm}}$, where $p_{tot,hs}$ is the

total gas pressure in the vial before the GC measurement).

### 2.5.4. Low molecular weight organic acids analysis

Acetic acid, formic acid, oxalic acid and pyruvic acid were quantified via ion chromatography (DX-320, Thermo Scientific, Sunnyvale, CA, USA). The instrument was equipped with an EG40 eluent gradient generator, a Dionex Ion Pack AG11-HC



RFIC 4 mm column and guard column, a Dionex AERS 500 4 mm electric suppressor and an electrical conductivity detector. Injection volume and flow rate were 250 µL and 1.5 mL min⁻¹, respectively. The following KOH gradient was used: $0 - 11$ min, 1 mmol L⁻¹; $11 - 37$ min, 1 mmol L⁻¹ to 40 mmol L⁻¹; 37 to 38 min, 40 mmol L⁻¹; 38 to 41 min, 1 mmol L⁻¹. In these conditions, acetic acid, formic acid, pyruvic acid and oxalic acid were eluted at 9.4, 11.9 13.0 and 24.9 min, respectively.

## 2.6. Cloud chamber instruments and techniques

### 2.6.1. Cloud condensation nuclei counter setup

The DOM samples were atomized by a home-made atomizer (based on a TSI aerosol generator), from a heart-shaped glass flask to minimize the amount of solution required for efficient aerosolization. The generated flow of polydispersed wet aerosols was then dried through orange silica gel and 4 Å molecular sieves; the exiting flow had a relative humidity < 5%. The

polydisperse flow of dry aerosols was used to generate a monodisperse flow with a TSI differential mobility analyser (DMA; TSI Model 3082) operating with a sheath to sample flow ratio of 10:1. The monodispersed dry aerosol flow was subsequently split in two and sampled by a TSI condensation particle counter (CPC-3772) and a DMT cloud condensation nuclei chamber (CCNC-100) (Roberts and Nenes, 2005; Rose et al., 2008). The CCNC operated with a total flow of 0.5 Lpm, corresponding to a high RH exposure time of approximately 10 s for the aerosols. We do not expect dynamic surface tension changes on this

time scale (Nozière et al., 2014). The number of activated droplets measured by the CCNC is then divided by the total number of particles measured by the CPC to yield the activated fraction and supersaturation (SS) curves. Specifically, the CCNC was operated by generating aerosol particles of a fixed dry diameter (80 nm) while changing supersaturations (SS) (0.2 % to 1.0 %) as well as using a fixed SS (0.40 %) while changing diameters (20 nm to 120 nm) to ensure reproducible measurements. Both methods were used to quantify the ability of DOM to act as a CCN by determining the κ values (Figure 1). The resulting

kappa values were used for the variance depicted in Figure 1, as triplicates were not conducted for every single time point because of limited samples. From Eq. (5) of the supercritical saturation ($S_c$), the hygroscopicity parameter κ is obtained following Eq. (6) (Petters and Kreidenweis, 2007; Ruehl et al., 2016):

$$S_c = \frac{D^3 - D_d^3}{D^3 - D_d^3(1 - \kappa)} exp\left(\frac{4\sigma_{s/a} M_w}{RT \rho_w D}\right) \qquad (5)$$

$$\kappa = \frac{4}{27 D_d^3 (ln S_c)^2}\left(\frac{4\sigma_{s/a} M_w}{RT \rho_w}\right)^3 \qquad (6)$$

where $S_c$ is the supercritical saturation, D is the wet diameter (not measured in this work), $D_d$ is the dry diameter (measured in this work by the DMA), κ is the hygroscopicity parameter (unitless), $\sigma_{s/a}$ is the surface tension of water (0.072 J m⁻²), $M_w$ is the molecular weight of water (18 g mol⁻¹), R is the universal gas constant (8.314 J K⁻¹ mol⁻¹), T is the temperature of the inlet flow (298 K) and $\rho_w$ is the density of water (10⁶ g m⁻³) (Petters and Kreidenweis, 2007).



The mass of the DOM samples are > 90% organic carbon by molar concentration and thus phase separation between inorganic ions and the organic phase is not likely occurring during aerosolization of the DOM. In other words, there is no thin film being generated, and thus the $\delta_{org}$ parameter, or the thickness of the organic film surrounding an inorganic core, is not calculated.

Two CCN and INP experiments performed one year apart showed reproducible data, indicating little degradation of the cloud droplet and ice nucleating activity due to storage.

### 2.6.2. Ice nucleation setup

The DRoplet Ice Nuclei Counter Zurich (DRINCZ) is a custom-built instrument that enables the quantification of heterogeneous ice nucleation through immersion freezing (David et al., 2019). Briefly, DRINCZ is an immersion freezing
technique with an optical detection employing in a cooling bath (Lauda ProLine RP 845, Lauda-Königshofen, Germany), a visible light camera (Microsoft Lifecam HD-3000) and a 96-well sterilized and sealed plate (PCR tray, 732-2386, VWR, USA). Each well of the plate is filled with 50 µL of the DOM solution using a multi-pipette with disposable tips. Above the cooling bath, a custom-machined aluminium plate holder keeps the plate in place so that the liquid in the wells is exactly level with the ethanol coolant. Additionally, a solenoid valve, triggered by a fluid level sensor, allows for additional ethanol, pre-cooled
to 0 °C, to be poured into the bath to compensate for the solvent's changing density with cooling and to ensure a constant ethanol level submerging the plate. A thick acrylic plate on top of the well plate weighs it down and prevents buckling of the plate to ensure that all the wells are immersed equally and experience the same temperature. The bath temperature is cooled at a rate of 1 °C min⁻¹. A diffuse LED array submerged at the bottom of the ethanol bath illuminates the plate from below and does not interfere with the coolant flow. The camera is mounted above the plate and captures the light transmission through
the wells every 15 s. The position of each well on the image is later automatically obtained through image recognition code in MATLAB and the change in light intensity for each well is obtained per image. Freezing fractions are then obtained for each well by taking the temperature at the time the image was taken and the instant when 60% of greatest change in light intensity occurred.

The DRINCZ measurement technique is limited to samples that freeze at temperatures warmer than −22.5 °C, since 50% of the wells of the background molecular biology reagent water (89079-460, Sigma Aldrich, USA) freeze at that temperature ($T_{50}$ = −22.5 °C). This detection limit of homogeneous freezing is largely due to well plate characteristics, which do not change for different well volumes, to water quality (Polen et al., 2018) and to a sterile working environment. While others have noted degradation of INP activity with sample storage (Stopelli et al., 2014), we did not observe a storage effect on the INP properties,
likely because our DOM samples were filtered on site to 0.20 µm using sterile filters and stored at 4 °C. Following repeated Sigma Aldrich water runs on DRINCZ, a master water background curve was generated. Specifically, 10 curves were used to generated the background in Figure 2(a) and lead to an average standard deviation in the freezing temperature detection of 0.2



°C. The background was corrected following (Vali, 1971, 2019), where the nucleus concentration, Nuc(T), of the background is subtracted from that of the sample and then inverted to reconstruct the corrected FF following Eq. (7) (David et al., 2019).

$$Nuc(T) = \frac{-1}{V_{well}\Delta T} \ln\left(1 - \frac{\Delta N}{N(T)}\right) \qquad (7)$$

where $V_{well}$ is the aliquot volume of 50 µL, $\Delta T$ is the change in temperature between each recorded image, N(T) is the number of unfrozen aliquots at the beginning of a temperature step, $\Delta N$ is the number of aliquots that freeze during the temperature step, and in DRINCZ's case, between recorded images. All reported FF curves have been background corrected.

The INP per mg of carbon is the $n_m$ value and was calculated following the quantitative analysis Eq. (8) according to Vali (Vali, 1971).

$$n_m = -\frac{\ln(1-FF)}{TOC \times V_{well}} \qquad (8)$$

where FF is the frozen fraction (value between 0 and 1), TOC is the concentration of non-purgeable organic carbon (mg C L$^{-1}$) and $V_{well}$ is the aliquot volume (50 µL). The data analysis was coded in MATLAB and in Igor Pro.

## 3. Results and discussion

### 3.1. Photochemical aging of DOM

To study the effect of photochemistry on CCN and INP, we sampled naturally occurring DOM from surface waters from the Dismal Swamp in Virginia, USA in 2014 and in 2016 and at the Suwannee River, Florida, USA in 2017. The natural DOM sampled from freshwater bodies is representative of a fraction of atmospheric organic aerosols directly relevant to lake spray aerosols and potentially prevalent in the aerosol phase over large fresh water bodies such as the Laurentian Great Lakes (Axson et al., 2016; Slade et al., 2010), the African Great Lakes, and Lake Baikal. Furthermore, we used commercially available Suwannee River fulvic acid (SRFA), an established standard for complex organic matter and humic-like substances in both aquatic (Cory et al., 2009; Guerard et al., 2009) and atmospheric chemistry experiments (Dinar et al., 2006; Frosch et al., 2011; Kessler et al., 2012; Ladino et al., 2016; Slade et al., 2017; Svenningsson et al., 2006; Wang and Knopf, 2011; Zelenay et al., 2011). To further bridge the gaps of aquatic and atmospheric chemistry, we used the same concentrations typically found in rivers and in cloud water, that is 20 mg L$^{-1}$ (Cook et al., 2017; Johannesson et al., 2004). Organic aerosols containing humic-like substances may have lower average molecular weights and lower aromatic moiety content that SRFA standards (Graber and Rudich, 2006), yet we complement our study with real surface collected waters to circumvent IHSS extraction and processing of DOM. Ultimately, we argue that our DOM samples naturally occurring in fresh water bodies and are adequate and alternative proxies for visualising the photomineralization mechanism identified in this study.

Bulk DOM samples were irradiated for 25 h with UVB light in a photoreactor in order to simulate the photooxidation processes occurring in the atmosphere. Using a combination of chemical actinometry and radiometry measurements, we estimate that



our 25 h-exposure experiments in the photoreactor are equivalent to approximately 55 hours of sunlight irradiation, further equivalent to 4.6 days in the environment. However, all experiments were conducted at 30-32 ℃ and an extrapolation to colder and more relevant atmospheric temperatures would slow the rates of reactions of photochemical processes but would not affect the photon flux. Many photochemical processes have weak temperature dependences, and for instance hydrogen peroxide quantum yields are affected by a factor of 1.8 per 10 ℃ (Kieber et al., 2014). Thus, the exposure time corrected for lower temperature is relevant in the context of an approximately one-week aerosol lifetime in the atmosphere.

The bulk solutions of irradiated DOM were subsequently analysed for CCN and INP activity. In all of our experiments, we employed concentrations between 14 and 20 mg of carbon L$^{-1}$ (mg C L$^{-1}$), equivalent to 1160 to 1670 μM of organic carbon, and represents typical aquatic DOM conditions (Johannesson et al., 2004) as well as typical concentrations found in cloud water (Cook et al., 2017). An important caveat to highlight here is that our photochemical experiments were conducted in bulk. It is thus likely that the effect quantified does not necessarily equate to identical processes in the droplet and/or in the organic aerosol phase. Yet, we expect the photoproducts and mechanisms within the bulk to be occurring within an aerosol droplet, potentially at faster rates because of higher organic concentrations in the aqueous aerosol versus in the bulk.

### 3.2. Photochemistry affects the hygroscopicity of DOM

To assess the ability of DOM to act as CCN, the $\kappa$ values of each irradiated DOM sample was calculated, where higher $\kappa$ values indicate a more effective CCN. During irradiation, we observed a clear increase in $\kappa$ values, up to 2.5 times higher than the non-irradiated samples (Figure 1). The change in CCN activity with increased UVB exposure was systematically observed across all of the naturally occurring samples tested, including the SRFA standard. In this study, $\kappa$ values for photochemically exposed DOM ranged between 0.12 and 0.45 with an instrumental error of 0.01 and an experimental uncertainty of less than 15%. These values compare to global ambient organic aerosol which have $\kappa$ values between 0.15 and 0.30 (Schmale et al., 2017).

In particular, SRFA's CCN activity has been reported to have $\kappa$ values of 0.06-0.08 (Dinar et al., 2006; Slade et al., 2017). In our experimental setup, SRFA has a $\kappa$ of $0.12 \pm 0.02$ and we attribute part of this discrepancy to the lower concentrations and the complete solubility of 20 mg C L$^{-1}$ of SRFA used in our experiments. Concentrations above 5000 mg C L$^{-1}$ were previously used and such concentrations form a suspension rather than a homogeneously dissolved solution potentially leading to lower κ (Dinar et al., 2006; Slade et al., 2017). This possible concentration effect on CCN activity could arise from the interfacial molecular arrangement of the organic polymeric material (Ruehl et al., 2016) which leads to changes in surface tension and thus to $\kappa$ values (Nozière et al., 2014; Ovadnevaite et al., 2017).




### 3.3. Photochemistry affects the ice nucleating ability of DOM

To probe the impact of photochemistry on ice nucleation, the field-collected DOM and SRFA were measured in our home-built DRoplet Ice Nuclei Counter Zurich (DRINCZ) (David et al., 2019). We report median freezing temperature ($T_{50}$) values which corresponds to the temperature at which 50% of the wells filled with a DOM solution froze (Figure 2(a)). The natural

DOM samples from different locations and different years show frozen fractions (FFs) significantly above the background ultra-pure water, demonstrating ice nucleation activity (Figure 2(a)). Specifically, DS 2014, DS 2016, SR 2017 and SRFA showed $T_{50}$ values of $-12.4 \pm 0.1$ °C, $-7.8 \pm 0.3$ °C, $-9.4 \pm 0.1$ °C and $-13.0 \pm 0.3$ °C, respectively, which is well above the $T_{50}$ value of ultra-pure water in DRINCZ (-22.5 °C). The steep curve of the FF curves exhibited by the samples suggest that the nature of ice nucleating material or active site is relatively homogeneous. These freezing temperatures are consistent with

recently identified river nanoscale INPs (Knackstedt et al., 2018; Moffett et al., 2018). The FFs were then normalized by the organic carbon concentration in the solution measured by a TOC analyser to yield $n_m$ values as a function of temperature (Fig. S1). These $n_m$ values physically describe the ice nucleation active mass in units of number per mg of carbon. We compare our $n_m$ values with those of the Wilson 2015 parameterization which represents the ice nucleating ability in immersion freezing mode of primarily organic material in the sea surface microlayer (Wilson et al., 2015) (Fig. S1). This comparison suggests that

our DOM samples, which are smaller than 0.20 μm, could be accounting for a subset of the IN activity previously observed in the sea surface microlayer (Irish et al., 2018; McCluskey et al., 2018; Wilson et al., 2015). Furthermore, sea spray aerosols have recently been shown to be effective INPs when enriched in organics (DeMott et al., 2016; Ladino et al., 2016; Si et al., 2018; Wilson et al., 2015), consistent with our results. Soil organic matter, also containing humic-like substances, is also able to act as INP in the immersion freezing mode (Hill et al., 2016; O'Sullivan et al., 2014). The SRFA was observed to have

lower IN activity than the collected DOM samples, despite SR 2017 having been collected near the IHSS collection point at Suwannee River. This result suggests that extraction and processing of Suwannee River DOM to yield its fulvic acid component reduces its IN activity, potentially related to differences in organic carbon content (Botero et al., 2018). In other words, the fulvic acid is not as good as an INP as other components within DOM. Our results for SRFA are consistent with heterogeneous ice nucleation studies which used aerosolized SRFA in deposition freezing mode and found the organic material

to be active (Wang and Knopf, 2011). Ultimately, DOM has the potential to nucleate ice at low to moderate ($-4$ to $-12$ °C) droplet supercooling through immersion freezing.

The DOM samples were then subjected to photochemical exposure in an identical set up as for the CCN experiments and the irradiated samples were measured on DRINCZ. Photochemistry decreased $T_{50}$ values as a function of UVB irradiation, with

an important impact on corresponding $n_m$ values (Figure 2). Indeed, the 2 to 3 °C change in the $T_{50}$ temperature observed in Figure 2(b) leads to corresponding changes of up to two orders of magnitude in $n_m$ values between the 0 h and the 25 h time points in Figure 2(c). In particular, for an $n_m$ of 100 ice nucleation sites per mg of carbon, DS 2016 experienced a suppression in IN activity of up to 4 degrees (Figure 2(c)). The irradiated DOM samples led to different spreads of $n_m$ values, indicating



DOM's composition-specific photochemistry response on immersion freezing and warrants further study. Furthermore, the rate loss of IN activity measured as $T_{50}$ could be quantified as a function of hours of UVB exposure. The slopes for photochemically exposed DS 2014, DS 2016, SR 2017 and SRFA were $-0.054$, $-0.145$, $-0.066$ and $-0.100$ $°C_{T50}$ $h^{-1}$, respectively (Figure 2(b)). The average decrease in $T_{50}$ values for irradiated DOM samples is thus $-0.09$ $°C_{T50}$ $h^{-1}$ of UVB light

and equates to $-0.04$ $°C_{T50}$ $h^{-1}$ of sunlight at ground level in mid-latitudes (see methods for conversion of photochemical reactor exposure to sunlight exposure). These values could be useful to the atmospheric modelling community to investigate a lifetime-dependent IN activity of organic matter such as DOM or sea spray aerosols.

Although our study is the first to quantify the effect of photochemistry on the ice nucleating ability of organic matter, others

have identified specific organic molecules and components that could help explain the IN ability of DOM. Wilson et al. parameterized the ice nucleating ability of sea surface microlayer samples using TOC and temperature (Wilson et al., 2015). However, this parameterization leads to higher estimates of $n_m$ than obtained for our DOM samples (Figure 2(c)). This result suggests that $n_m$ does not efficiently constrain the IN activity of DOM. McCluskey et al. arrived at a similar conclusion for the IN activity of sea spray aerosol (McCluskey et al., 2018). The INPs measured in this study were soluble with an operational

definition of passing through a 0.2 μm filter and could account for a subset of INPs identified in marine samples. In addition, soluble INPs have been shown to contain extracts from plant-based material, including cellulose (Hiranuma et al., 2015) and proteinaceous material (Augustin et al., 2013; Dreischmeier et al., 2017; Koop and Zobrist, 2009; O'Sullivan et al., 2015; Pummer et al., 2012, 2015; Wilson et al., 2015), which we know are present within our DOM samples. Finally, Gute and Abbatt observed that OH radical oxidation of pollen in deposition freezing required higher supersaturation ratio with respect

to ice (Gute and Abbatt, 2018). Our results nicely corroborate with this observation and further suggests that not only OH radicals but photochemical processes in general are able to decrease the ice nucleation ability of organics, including pollen, in more than one freezing mode.

### 3.4. Chemical changes occur due to photochemistry

To understand the molecular origin of the observed photochemical impacts of DOM on cloud droplet (Figure 1) and ice

nucleation (Figure 2), we tracked total carbon, absorbance, conductivity, pH, organic acids, CO and $CO_2$ during irradiation.

### 3.4.1. CO and $CO_2$

The photooxidation products CO and $CO_2$ were quantified by gas chromatography with flame ionization detector (GC-FID) equipped with a methanizer between the GC column and the FID detector. The major and final product was $CO_2$, accounting for 50 to 100 % of dissolved organic carbon mass loss (Figure 3). On the other hand, CO accounted for no more than 0.1 %

mass yield, indicating that the original organic carbon content of DOM is removed primarily by photomineralization to $CO_2$ after 25 h of UVB irradiation (Figure 3). It must be noted that the CO and $CO_2$ concentrations reported here represent a lower





limit, as these particular irradiation experiments needed to be performed in a headspace-free setting to limit analyte loss. This setup consequently limits dissolved $O_2$ availability, potentially limiting the extent of oxidation. However, this limitation is not expected to occur in liquid atmospheric aerosols. The $CO_2$ yields presented in this study serve as experimental evidence for the ultimate fate of the organic carbon aerosol pool: the production of $CO_2$.

### 3.4.2. Organic acids and pH

Formic, acetic, oxalic and pyruvic acid are known DOM photooxidation products (Moran and Zepp, 1997) and have been previously quantified in organic aerosols, further supporting their identification as tracers for photooxidation (Boreddy et al., 2017; Pillar and Guzman, 2017; Zhang et al., 2016). All four organic acids were produced during the photooxidation of all the DOM samples (Figure 3). In addition, oxalate has been recently quantified to be a major photoproduct of dissolved organic carbon from primary wildfire emissions (Tomaz et al., 2018). Formic acid and acetic acid concentrations increase continuously, but oxalic acid and pyruvic acid concentrations show a growth-and-decay profile (Figure 3). Indeed, pyruvic acid is a known intermediate photoproduct on the pathway to the mineralization products of $CO_2$ (Griffith et al., 2013).

The quantified production of formic, acetic, oxalic and pyruvic acids supports the changing chemistry observed during aqueous photooxidation related to the changes in CCN; high O/C ratio leads to higher $\kappa$ values. It is clear that the same oxidation products are produced and thus the same photomineralization mechanism is operating in all DOM samples, albeit to different extents. This difference is attributed to the ability of the DOM samples to absorb irradiation and to generate different amounts of reactive oxygen species. These acids also do not appear to increase the efficiency of DOM to act as an INP, suggesting a significant role of the larger molecules within DOM to nucleate ice by immersion freezing.

In addition, the pH of the solutions was measured to be approximately five at each time point for the field-collected DOM samples and remained unchanged throughout the irradiation period. The pH values further support the absence of carbonate anions in solution.

### 3.4.3. Total carbon and conductivity

All DOM solutions reproducibly lost organic carbon during UVB exposure (Figure 4). The organic carbon loss was dependent on the DOM sample and ranged between 15 % (SRFA) and 63 % (DS 2014) over the course of 25 h of UVB exposure (Figure 4). Specifically, the $\kappa$ value of SRFA upon 25 h of UVB exposure increased the least compared to the natural DOM sample, and can be explained by changes in TOC. Furthermore, the inorganic carbon fraction was quantified and found to be consistently below 0.3 mg C L-1, indicating no accumulation of carbonate in solution. The conductivity of the DOM samples remained constant over UVB exposure at $60 \pm 14$ $\mu$S cm-1, $19 \pm 1$ $\mu$S cm-1, $28 \pm 1$ $\mu$S cm-1 and 30 $\mu$S cm-1 for DS 2014, DS 2016, SR 2017 and SRFA, respectively, indicating no meaningful change in ionic species concentration. The conductivity



measurements rule out the hypothesis that the increase in CCN and the decrease in INP during irradiation is due to the production of ionic species.

In our DOM samples, there are insufficient inorganic ions for the organic matter to act as an organic film when aerosolized. Yet, photomineralization leads to a decrease in the organic carbon and a decrease in the organic-to-inorganic ratio (Figure 5). The decreasing organic carbon concentrations at the surface of the droplet may be increasing the surface tension of the particle, bringing it closer to the surface tension of water (Kristensen et al., 2014; Petters and Petters, 2016). Remarkably, decreases of organic carbon affect all DOM in the exact same way, evidenced by the identical slopes of $-0.02$ (per mg C L$^{-1}$) of all $\kappa$ vs organic carbon in Figure 5. This result supports the notion that the change in CCN activity is dependent on the decrease in organic carbon content. Furthermore, the smallest change in $\kappa$ exhibited by SRFA is correlated to the smallest change in its organic carbon content in comparison to the natural DOM samples, indicating a non-negligible impact of processing on the commercial sample.

Finally, the change in CCN and INP was correlated with organic carbon content (Figure 5). The decrease in INP ability due to photochemistry (Figure 5) suggests that the organic carbon is likely the source of ice nucleation activity. In contrast to the changes in CCN ability, which all demonstrated identical dependence on TOC (see Figure 5), the INP ability dependence on TOC was variable. Interestingly, Dismal Swamp DOM 2014 experienced the highest organic carbon conversion (Figure 4), yet its decreasing T50 values lead to a negligible impact on its nm values (Figure 2(c)). Although chromophoric species appear to be key INP contributors, we conclude that the chromophoric species alone in this specific DOM sample were not solely responsible for the IN activity. Specific structures or chemical moieties of the organic content responsible for the ice nucleation activity still elude us (Knopf et al., 2018). On the other hand, the irradiated Dismal Swamp DOM 2016 sample experienced the most pronounced change in $T_{50}$ values and an intermediate change in organic carbon content which also led to a large decrease in $n_m$ values. Figure 5 highlights the different linear slopes observed between $T_{50}$ values and organic carbon and suggests that total organic carbon alone is not an accurate parameter for predicting INP. Rather, the organic carbon is photomineralized but the ice-active component of the DOM is being affected to different extents within the different samples. This result suggests that specific chemical moieties could be responsible for the immersion freezing activity.

### 3.4.4. Absorbance and photobleaching

The DOM material is coloured due to the presence of chromophoric components within the natural samples. Upon photooxidation, the sample absorbance decreases as a function of UVB exposure, and photobleaching is observed at all wavelengths (Figure 6). This photobleaching process has also been experimentally characterized for aerosol probe molecules and for biomass burning aerosols (Wong et al., 2017; Zhao et al., 2015). We show here that photobleaching is concurrent with a gain in CCN, a loss in IN activity and a loss in organic carbon. These results indicate that the chromophoric species within all the DOM samples are being photomineralized with UVB exposure and this mechanism is expected to take place within the



organic aerosol one-week lifetime in the atmosphere. Our experiment results are further corroborated by a recent modelling study which proposes an ageing scheme for photobleaching of organic aerosols to further increase the correlation between modelled and observed brown carbon absorption (Wang et al., 2018).

### 3.4.5. Photochemistry and the photomineralization mechanism

Upon irradiation of DOM, reactive intermediates such as triplet state DOM, OH radicals, singlet oxygen $^1O_2$, and peroxides can be formed (McNeill and Canonica, 2016; Rosario-Ortiz and Canonica, 2016). The reactive oxygen species are produced as a result of indirect photochemical processes involving the chromophoric components of DOM, which act as photosensitizers within organic aerosols (Arangio et al., 2016; Corral Arroyo et al., 2018; Laskin et al., 2015). The combination of both direct photodegradation and oxidation by reactive intermediates leads to the photomineralization of the organic carbon present in

DOM. This process is ubiquitous in natural aquatic systems, and it represents a competitive form of dissolved organic carbon processing in surface environments such as arctic lakes and rivers along with microbial processing (Miller and Zepp, 1995). It is therefore reasonable to assume that this process is operating in the photochemical aging of aqueous organic aerosols. Nevertheless, a detailed, arrow-pushing chemical mechanism to get from complex organic matter to organic acids to $CO_2$ remains elusive.

These photochemical processes are occurring in situ and are not due to external heterogeneous oxidation. In fact, SRFA heterogeneous oxidation by OH radicals in a flow tube experiment does not lead to substantial organic carbon mass loss (Kessler et al., 2012), indicating the significance of in situ photochemistry compared to external gas-phase heterogeneous oxidation. Indeed, we show that photochemical processing can be a dominant atmospheric aging process, impacting CCN and

INP efficiencies and concentrations.

## 4.    Atmospheric implications

Models of cloud microphysics often use ground based CCN and INP measurements for validation studies. Regional models and GCMs which contain prescribed CCN concentrations may be underestimating their contribution to clouds, if the input is based on data collected and measured at ground level close to emission sources. The global average of $\kappa$ is estimated to be 0.3

based on a recently harmonized CCN worldwide dataset (Schmale et al., 2017). If the CCN activity of the organic fraction of the aerosol population were to increase with altitude due to longer photochemical exposure, one should expect a higher CCN activity of the aerosol population. A change of $\kappa$ values from 0.15 to 0.35, as observed in our own results for DOM, for a simulated particle of a 100 nm dry diameter, could change the supercritical saturation conditions for 50% CCN activated fraction from 0.30 SS% to 0.20 SS% (above 100%). The impact of photochemical ageing would then depend on the aerosol

size distribution, but with a typical accumulation mode distribution reported by Paramonov et al. (Paramonov et al., 2015), a decrease of required critical supersaturation from 0.30 SS% to 0.20 SS% to activate the same population of aerosols could




increase the activated fraction from 0.3 to 0.4. This change would lead to an earlier onset and a higher concentration of cloud droplets in a lifting air mass, thus forming clouds at lower altitudes with potential effects on the brightness and optical depth of the cloud. Changes in particle size during photochemistry could in turn affect the CCN activation conditions, however only small reductions in particle diameter have been previously reported (up to 4%) for chemical ageing reactions relevant to the
atmosphere (George et al., 2007).

Whether is it is a dust particle covered in an organic coating, a bioaerosol or a secondary organic aerosol, many INPs contain organic carbon material (Knopf et al., 2018; Schmidt et al., 2017), and would thus be prone to photomineralization once in a droplet. The decrease in INP concentrations with photochemical processing results in a change in $T_{50}$ of up to 4 °C over a 25
h-UVB exposure corresponding to a simulated exposure time of 55 sunlight hours. This temperature change could impact the ratio of ice to water droplets within a mixed-phase cloud by delaying the onset of glaciation and increasing the supercooled liquid fraction of the cloud due to the lower $T_{50}$ temperatures, thus modifying the radiative properties and the lifetime of the cloud. If the decrease in IN activity due to photochemical ageing is omitted from models, ice formation may be occurring at higher temperatures, leading to an earlier more effective glaciation of mixed-phase clouds than in reality, which has been
reported to occur in models (Lohmann and Neubauer, 2018). Earlier glaciation has implications for cloud lifetime and precipitation, leading to an underestimation of the importance of the cloud phase feedback in anthropogenic climate change (Tan and Storelvmo, 2015). Indeed, it has been shown that accurate representation of supercooled liquid fraction is crucial to predicting climate sensitivity in a future warming climate (Tan et al., 2016).

**Author contributions**

NBD, ZAK and KM designed the experiments with contributions from ROD and RO. NBD, RO, LSB, ROD and VW conducted the experiments and collected data. NBD analysed the data. Manuscript was written by NBD with contributions from ZAK, RO, ROD and KM.

**Acknowledgments**

We acknowledge Prof. Kenneth Mopper, Dr. Vivian Lin and Dr. Paul Erickson for the field collection of the dissolved organic
matter water in 2014, 2016 and 2017 respectively. We acknowledge technical help from Dr. Björn Studer and Dr. Martin Schroth with the TOC analyser and the GC-FID, respectively.



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

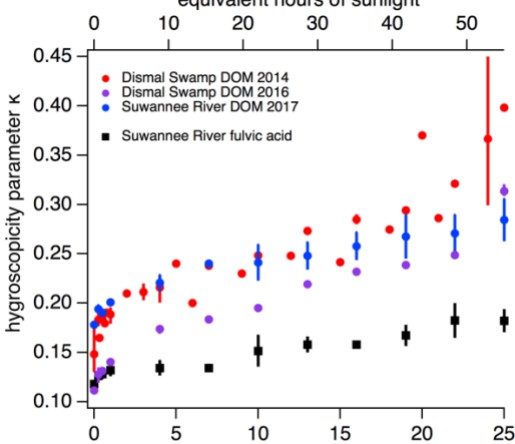

**Figure 1. The CCN ability of DOM and of SRFA increases as a function of irradiation. The experimental conditions are equivalent to up to 25 h of UVB exposure (~55 hours of sunlight in the atmosphere). The colors indicate different source material (DS 2014 in red, DS 2016 in purple, SR 2017 in blue and SRFA in black) and the symbols indicate the material was field-collected (circles) or commercially available (squares). The instrumental uncertainty in $\kappa = \pm 0.01$. To show the spread of the measurements, the variance from multiple independent experiments are illustrated as bars. The variance was used since not every point had triplicates for standard deviations to be calculated.**





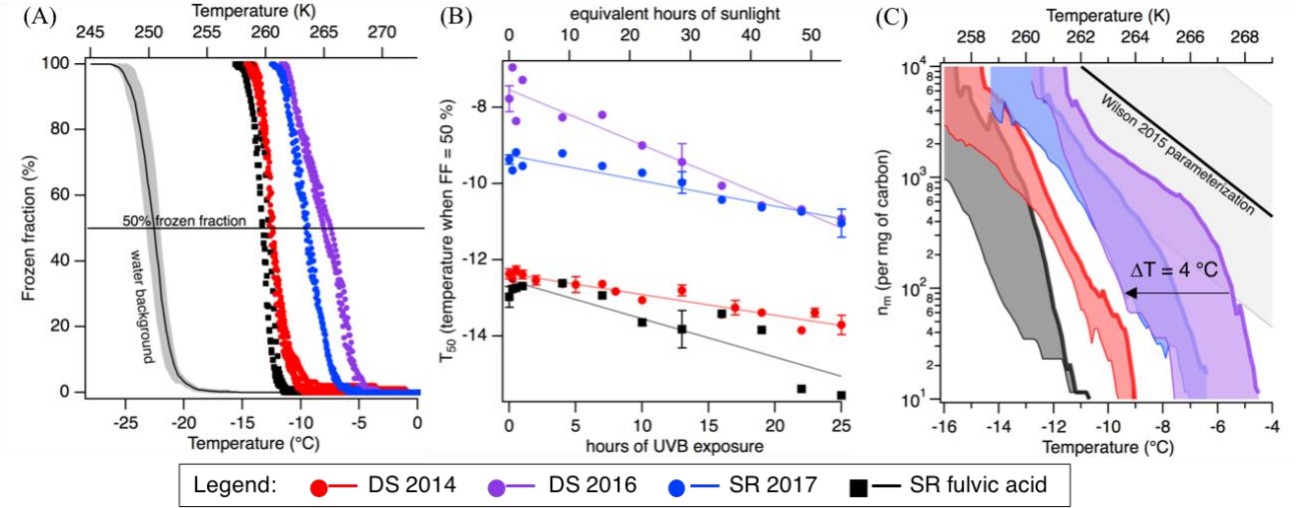

**Figure 2. DOM is IN active. (A)** The frozen fractions (FF) as a function of temperature are displayed for field-collected DOM from the Dismal Swamp, Virginia, USA in 2014 (DS 2014, red) and in 2016 (DS 2016, purple), from the Suwannee River, Florida, USA in 2017 (SR 2017, blue), and for the reference material Suwannee River fulvic acid (SR fulvic acid, black). The solid 50% frozen fraction line is used to determine the $T_{50}$, the temperature at which 50% of the droplets are frozen. Specifically, DS 2014, DS 2016, SR 2017 and SRFA showed $T_{50}$ values of –12.4 ± 0.1 °C, –7.8 ± 0.3 °C, –9.4 ± 0.1 °C and –13.0 ± 0.3 °C, respectively. The black line represents the water background FF curve with standard deviation (1 σ) from 10 different water background run over a period of 6 months. **(B)** $T_{50}$ as a function of UVB exposure for all DOM samples is shown. All samples exhibit a decrease in IN activity upon 25 h of UVB exposure (~ 55 hours of sunlight in the atmosphere). The lines are linear fits to the data to quantify the slope and the rate loss of IN activity. The slopes are –0.054, –0.145, –0.066 and –0.100 for DS 2014, DS 2016, SR 2017 and SRFA, respectively. **(C)** The shaded regions in the $n_m$ space correspond to the extent of change in IN activity due to photochemistry from the mean values at t = 0 h to t = 25 h. Note that the $n_m$ values were determined by accounting for the change in organic carbon content measured by TOC analyzed and depicted in Figure 4. The Wilson 2015 parameterization for $n_m$ of INPs in the sea surface microlayer is also shown for comparison (Wilson et al., 2015).





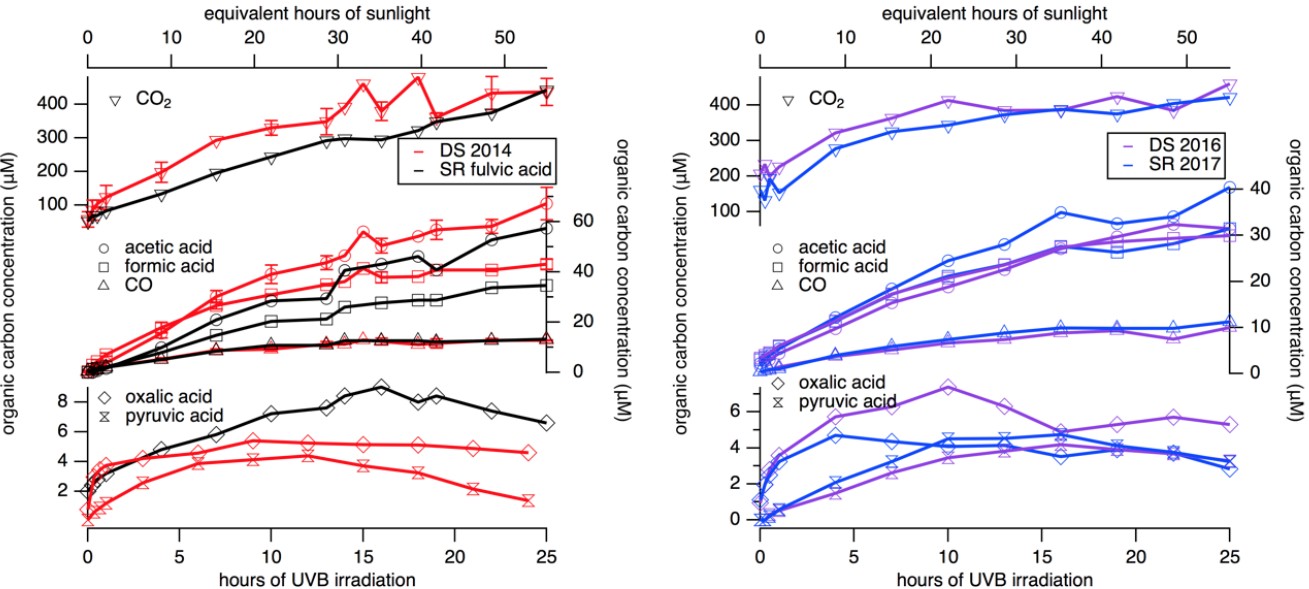

**Figure 3. Photooxidation products are formed from DOM. This graph depicts data from (a) DS 2014 and SRFA and from (b) DS 2016 and SR 2017. $CO_2$ and CO were quantified via GC-FID, whereas formic, acetic, oxalic and pyruvic acids were quantified by**
5 **ion chromatography. Pyruvic acid for SRFA was below detection limits. Note that the concentrations are expressed per moles of carbon and so organic acids with multiple carbon atoms have been normalized.**

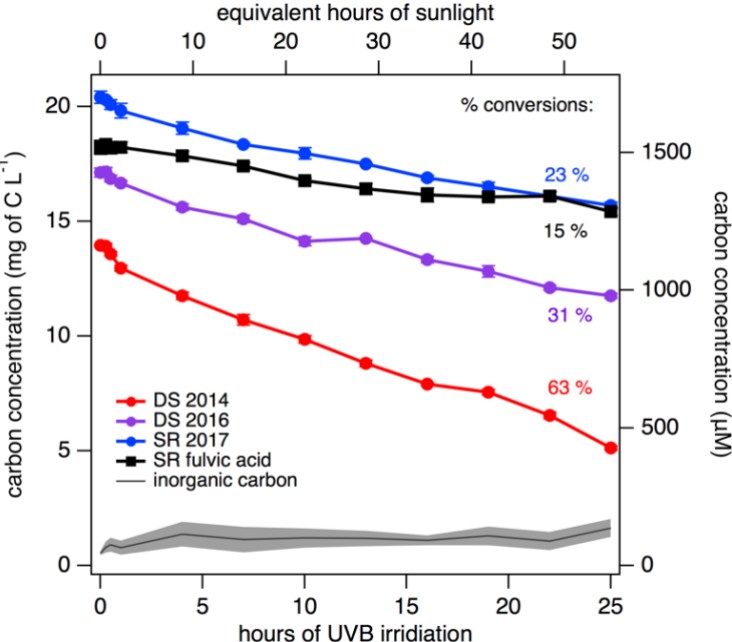

**Figure 4. Decrease of the dissolved organic carbon (in both mg C L$^{-1}$ and in µM) present in DOM upon UVB irradiation. Both the organic (as non-purgeable organic carbon) and inorganic carbon (as carbonates) were quantified by a TOC analyzer, which was**





calibrated using standard solutions of terephthalic acid and carbonate, respectively. Note that dissolved $CO_2$ is not quantified by TOC analysis. The inorganic carbon is averaged for all samples with the standard deviation in grey. The percentage values indicate the conversions of original organic matter into photoproducts.

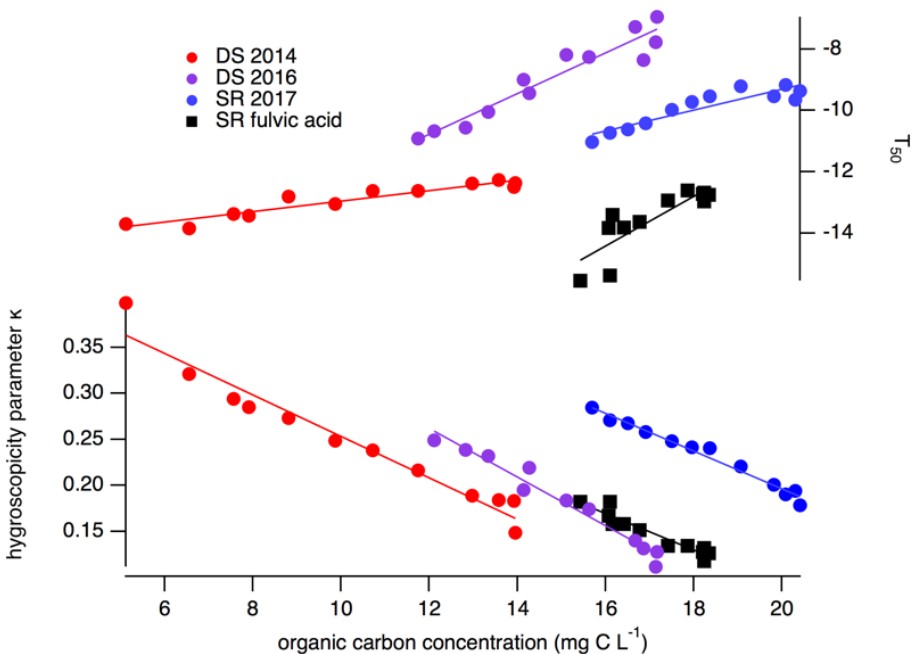

**Figure 5.** CCN abilities and IN abilities of DOM as a function of organic carbon. Slopes for $\kappa$ vs organic carbon are all $y = -0.02\ x$. However, the slopes for $T_{50}$ vs organic carbon are different for different DOM: 0.17, 0.66, 0.34 and 0.81 for DS 2014, DS 2016, SR 2017 and SRFA, respectively.





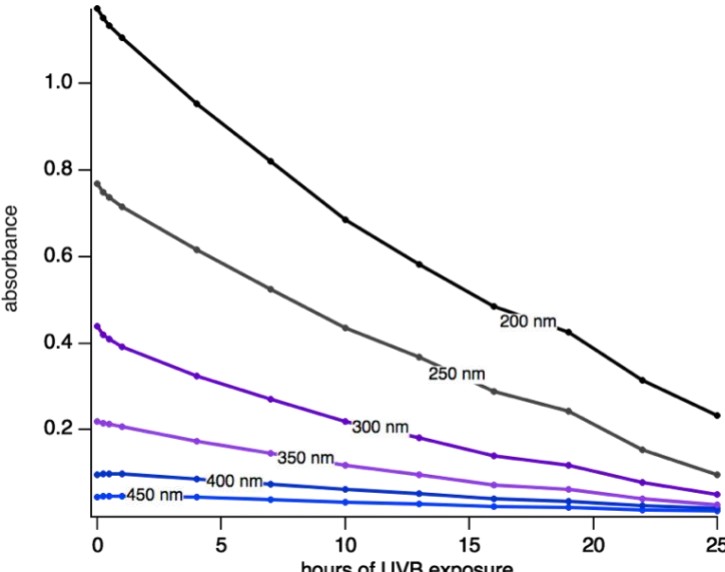

**Figure 6: DOM absorbance decreases as a function of irradiation. Absorbance of a 14 mg C L$^{-1}$ of Dismal Swamp 2014 (DS 2014) as a function of UVB exposure leads to photobleaching of the DOM.**