# Peer review of "Photomineralization mechanism changes the ability of dissolved organic matter to activate cloud droplets and to nucleate ice crystals"

_Atmospheric Chemistry and Physics, 2019_

## Referee Comment (RC1) · Anonymous Referee #1 · 19 Jun 2019

This manuscript describes a study aimed at determining how photochemistry of water soluble organic matter in aerosol droplets affects the ability of these aerosols to be cloud condensation (CCN) or ice nuclei (INP). Natural organic matter collected directly from rivers in the Southeastern U.S. and a fulvic acid isolate from the international humic substances society (IHSS) were photolyzed for a length of time approximately equivalent to the lifetime of an aerosol droplet in the atmosphere. The solutions were atomized into a cloud condensation nuclei chamber to determine their kappa-values (measure of hygroscopicity from Koehler theory) or studied for their ice nucleating abil-

ity in an immersion freezing apparatus. The authors found that irradiation of solutions led to significant mineralization of carbon (loss of CO and CO2) and formation of small carboxylic acids due to fragmentation induced by direct and indirect photochemistry promoted by chromophores within the dissolved organic matter; total dissolved organic carbon decreased over time of photolysis. Interestingly, hygroscopicity values increases with decreasing dissolved organic matter concentrations while INP efficiencies decreased. This suggests that photochemical aging makes aerosols better CCN and worse INP. The data provides valuable information that can be used to model aerosol/cloud processes and their corresponding radiative properties.

The manuscript is excellent in all respects, from the quality of the writing to the figures, and of course the science. Regarding the science, there are several aspects of this manuscript that make the work novel. Aside from the fact that the CCN and INP measurements were carried out using state of the art instrumentation, measuring both CCN and INP activity on the same types of samples is quite unique and allow critical insights to be made; I am not aware of another study that has done this. Another unique aspect of this work, which stems from the multidisciplinary approach of this work, is the use of dissolved organic matter collected from the Great Dismal Swamp in Virginia and the Suwannee River in Florida, USA. These are more realistic surrogates of dissolved organic matter since they were not subjected to the type of extraction protocol that the typical commercially available humics or fulvics are subjected to that artificially change their composition and reactivity relative to unaltered samples. Finally, another strength of the manuscript lays in the quality and quantity of supporting photochemical (actinometry) and analytical chemistry measurements (CO(g), CO2(g) loss and production of small organic acids) that help explain the trends in CCN and INP efficiencies.

The most significant findings are that in situ photochemistry driven by water soluble organic matter leads to significant organic carbon mass loss and this loss impacts CCN and INP efficiencies and concentrations. Indeed, the fact that the photochemistry occurred within the aerosol solutions and were not impacted by gas phase oxidants,

suggests that photochemistry from within the particle may be more important factors driving CCN and INP efficiencies than heterogeneous oxidation mechanisms involving gas-to-particle partitioning of oxidants. This will be a significant finding for the aerosol modeling community and I very much appreciated the clear discussion of the impacts of their work on predicted aerosol size distributions, aerosol properties, and the impact on cloud formation and optical properties. I feel the manuscript is ready for publication and I congratulate them on some very nice work.

---

## Referee Comment (RC2) · Anonymous Referee #2 · 2 Aug 2019

Overview

Borduas-Dedekind et al. report on systematic laboratory studies of the photo-degradation of representative dissolved organic matter bulk solutions and its impact on cloud condensation nuclei (CCN) and ice nuclei activity (IN). Their main finding is that photochemical oxidation promotes CCN activity and suppresses IN activity. A perhaps remarkable finding is that the hygroscopicity of these organic mixtures increases from ∼0.1 to upwards of ∼0.45 with increasing irradiation. This is where I feel the authors need to strengthen their argument - can such an increase be reconciled? The meth-

ods, techniques, and analyses appear well executed and cited, and the authors did an excellent job discussing caveats and experimental limitations. I support publication in Atmospheric Chemistry and Physics after the authors address my comments. In addition to addressing the large enhancement in $\kappa$, the paper could really be improved by including a schematic to connect the different chemical processes the authors propose are happening.

Comments

Abstract, line 13: Be careful with phrasing here – the authors are measuring the CCN and IN activity, not necessarily ability to form mixed-phase clouds. I suggest dropping "...ability to form mixed-phase clouds, by acting as..." and connect so that it reads, "...on its cloud condensation nuclei (CCN) and ice nuclei (IN) activity."

Page 7, line 25: Please clarify how the wet diameter was determined in the calculation of critical supersaturation if such data was not measured. Do you mean it was a fitting parameter as part of the activated fraction curve?

Page 10, lines 10-13: I much appreciate the authors' statement of caveats and potential differences in their bulk measurements compared to aerosol phase processes. Beyond the kinetic considerations due to differences in concentrations between the aerosol phase and bulk solution, other factors such as mixing state, phase, morphology (i.e., matrix effects) might also impact photochemical processes differently in aerosol compared to the bulk solution (Lignell et al., 2014).

Page 10, section 3.2: This discussion needs some revision – please describe in detail why $\kappa$ varies between 0.12 and 0.45. Would you expect $\kappa$ to increase this much, and why? This is significant given that $\kappa$ for organic aerosol is generally $\sim$0.1-0.2 (Petters and Kreidenweis, 2007). See also comment (6) below.

Page 13, line 25: What are the units referring to carbon loss? Is it by mass, volume, molar concentration?

It's difficult for me to make a clear connection between the increase in hygroscopicity and the production of small organic acids/loss of carbon, except that the author's state there is a dependency of CCN activity on the decrease in organic carbon content. Is the reasoning that the production of small organic acids increases average O/C of the particles and thus $\kappa$? Is there any evidence that shows such an enhancement in CCN activity attributed to an increase in O/C and how much of a change in O/C is expected? Perhaps the significant loss of carbon (up to 63%) during irradiation, likely due to volatilization, decreases average DOM molar mass, thus increasing $\kappa$, e.g., as in Slade et al. (2017) for Suwannee River Fulvic Acid particles reacting with OH? The following relationship relates molar mass to $\kappa$ (Mikhailov et al., 2013):

$\kappa$=J*(density/M)/(density_w/M_w )

Where J is the van't Hoff factor, M is average particle molar mass, and the subscript "w" refers to water. Assuming constant particle density and unity van't Hoff factor, a decrease in molar mass would increase hygroscopicity. I understand the authors did not analyze the mixtures using mass spectrometry, however, performing such a calculation relating $\kappa$ to molar mass could help support the argument even further.

Page 13, line 21: I found the result that pH of the solution remained at ∼5 during the irradiation quite surprising. If more organic acids are generated with increasing irradiation, would not the pH further decrease?

The paper could really benefit from an overall schematic of the proposed processes (mechanisms) occurring. For example, a reaction scheme that shows the production of oxidants internal of the solution, the oxidation of a representative DOM structure and formation of the small organic acids, etc. There are many (exciting and cross-cutting) points discussed in the paper. However, connecting them without a clear schematic, makes it very difficult to follow in the paper.

References

[Figure]

Lignell, H., Hinks, M. L., Nizkorodov, S. A.: Exploring matrix effects on photo-chemistry of organic aerosols, Proc. Natl. Acad. Sci. USA, 111, 13780-13785, 10.1073/pnas.1322106111, 2014.

Mikhailov, E., Vlasenko, S., Rose, D., and Poschl, U.: Mass-based hygroscopicity parameter interaction model and measurement of atmospheric aerosol water uptake, Atmos Chem Phys, 13, 717-740, 10.5194/acp-13-717-2013, 2013.

Petters, M. D., and Kreidenweis, S. M.: A single parameter representation of hygroscopic growth and cloud condensation nucleus activity, Atmos. Chem. Phys., 7, 1961-1971, 10.5194/acp-7-1961-2007, 2007.

Slade, J. H., Shiraiwa, M., Arangio, A. M., Su, H., Pöschl, U., Wang, J., and Knopf, D. A.: Cloud droplet activation through oxidation of organic aerosol influenced by temperature and particle phase state, Geophys. Res. Lett., 44, 10.1002/2016GL072424, 2017.

---

## Author Comment (AC1) · 21 Aug 2019

We thank the reviewer for supporting our work and extend a sincere thank you for their positive feedback and for their time in reading, reviewing and commenting on our manuscript.

---

## Author Response (AR1)

Referee #1:

This manuscript describes a study aimed at determining how photochemistry of water soluble organic matter in aerosol droplets affects the ability of these aerosols to be cloud condensation (CCN) or ice nuclei (INP). Natural organic matter collected directly from rivers in the Southeastern U.S. and a fulvic acid
isolate from the international humic substances society (IHSS) were photolyzed for a length of time approximately equivalent to the lifetime of an aerosol droplet in the atmosphere. The solutions were atomized into a cloud condensation nuclei chamber to determine their kappa-values (measure of hygroscopicity from Koehler theory) or studied for their ice nucleating ability in an immersion freezing apparatus. The authors found that irradiation of solutions led to significant mineralization of carbon (loss
of CO and CO2) and formation of small carboxylic acids due to fragmentation induced by direct and indirect photochemistry promoted by chromophores within the dissolved organic matter; total dissolved organic carbon decreased over time of photolysis. Interestingly, hygroscopicity values increases with decreasing dissolved organic matter concentrations while INP efficiencies decreased. This suggests that photochemical aging makes aerosols better CCN and worse INP. The data provides valuable information
that can be used to model aerosol/cloud processes and their corresponding radiative properties.

The manuscript is excellent in all respects, from the quality of the writing to the figures, and of course the science. Regarding the science, there are several aspects of this manuscript that make the work novel. Aside from the fact that the CCN and INP measurements were carried out using state of the art instrumentation, measuring both CCN and INP activity on the same types of samples is quite unique and
allow critical in- sights to be made; I am not aware of another study that has done this. Another unique aspect of this work, which stems from the multidisciplinary approach of this work, is the use of dissolved organic matter collected from the Great Dismal Swamp in Virginia and the Suwannee River in Florida, USA. These are more realistic surrogates of dissolved organic matter since they were not subjected to the type of extraction protocol that the typical commercially available humics or fulvics are subjected to that
artificially change their composition and reactivity relative to unaltered samples. Finally, another strength of the manuscript lays in the quality and quantity of supporting photochemical (actinometry) and analytical chemistry measurements (CO(g), CO2(g) loss and production of small organic acids) that help explain the trends in CCN and INP efficiencies.

We thank the review for supporting our work.

The most significant findings are that in situ photochemistry driven by water soluble organic matter leads to significant organic carbon mass loss and this loss impacts CCN and INP efficiencies and concentrations. Indeed, the fact that the photochemistry occurred within the aerosol solutions and were not impacted by gas phase oxidants, suggests that photochemistry from within the particle may be more important factors driving CCN and INP efficiencies than heterogeneous oxidation mechanisms involving gas-to-particle
partitioning of oxidants. This will be a significant finding for the aerosol modeling community and I very much appreciated the clear discussion of the impacts of their work on predicted aerosol size distributions, aerosol properties, and the impact on cloud formation and optical properties. I feel the manuscript is ready for publication and I congratulate them on some very nice work.

A sincere thank you for the reviewer's positive feedback,

Referee #2

Overview

Borduas-Dedekind et al. report on systematic laboratory studies of the photo- degradation of representative dissolved organic matter bulk solutions and its impact on cloud condensation nuclei (CCN)
and ice nuclei activity (IN). Their main finding is that photochemical oxidation promotes CCN activity and suppresses IN activity. A perhaps remarkable finding is that the hygroscopicity of these organic mixtures increases from ~0.1 to upwards of ~0.45 with increasing irradiation. This is where I feel the authors need to strengthen their argument - can such an increase be reconciled? The methods, techniques, and analyses appear well executed and cited, and the authors did an excellent job discussing caveats and experimental
limitations. I support publication in Atmospheric Chemistry and Physics after the authors address my comments.

We thank the reviewer for their feedback and address the comments individually below.

In addition to addressing the large enhancement in κ, the paper could really be improved by including a schematic to connect the different chemical processes the authors propose are happening.

ACP does not typically require a TOC graphic, but upon the request of the reviewer we have generated a new Figure 7 with the overall mechanism being described.

[Figure]

Figure 7: Overview of the impacts of the photomineralization mechanism on photochemical changes of DOM and on DOM-cloud
interactions. After field collected DOM and SRFA were exposed to the equivalent of 4.5 days of sunlight, the TOC concentrations decreased between 15 to 63% conversion, while the production of CO, $CO_2$ and organic acids, including formic acid, acetic acid, oxalic acid and pyruvic acid, increased. In addition, the absorbance of the material decreased with photooxidation leading to photobleaching. These physicochemical changes are linked to a decrease in IN ability and an increase in CCN ability.

Comments

Abstract, line 13: Be careful with phrasing here – the authors are measuring the CCN and IN activity, not necessarily ability to form mixed-phase clouds. I suggest dropping "…ability to form mixed-phase clouds, by acting as…" and connect so that it reads, "…on its cloud condensation nuclei (CCN) and ice nuclei (IN) activity."

Agreed. We amended the sentence so that it now reads, "*In this study, we quantify on a molecular scale*
*the effect of photochemical exposure of naturally occurring dissolved organic matter (DOM) and of a fulvic acid standard on its cloud condensation nuclei (CCN) and ice nucleation (IN) activity.*"

Page 7, line 25: Please clarify how the wet diameter was determined in the calculation of critical supersaturation if such data was not measured. Do you mean it was a fitting parameter as part of the activated fraction curve?

The reviewer is correct. $D_{wet}$ is a fitted parameter as part of the activated fraction when employing equation 6. The comment in parentheses now reads, *"(fitted parameter when using Eq. (6))"*.

Page 10, lines 10-13: I much appreciate the authors' statement of caveats and potential differences in their bulk measurements compared to aerosol phase processes. Beyond the kinetic considerations due to differences in concentrations between the aerosol phase and bulk solution, other factors such as mixing state, phase, morphology (i.e., matrix effects) might also impact photochemical processes differently in aerosol compared to the bulk solution (Lignell et al., 2014).

We thank the reviewer for this comment and additional caveats. We have kept our current discussion and added a sentence addressing aerosol physics vs DOM caveat with the recommended reference in addition to McDow's et al. seminal work. This sentence at the end of the paragraph reads, "In addition, organic aerosols can also exist under different mixing states, phases and morphologies likely leading to matrix effects which were not captured in this work (Lignell et al., 2014; McDow et al., 1996)."

Page 10, section 3.2: This discussion needs some revision – please describe in detail why κ varies between 0.12 and 0.45. Would you expect κ to increase this much, and why? This is significant given that κ for organic aerosol is generally ~0.1-0.2 (Petters and Kreidenweis, 2007). See also comment (6) below.

We thank the reviewer for the comments. We have amended the discussion in light of the additional point below. It now reads, "We further note the seemingly large $\kappa$ values for organic matter, specifically for the irradiated field collected samples DS 2014, DS 2016 and SR 2017 (Figure 1). We suggest that as the organic matter is photomineralized, the organic-to-inorganic ratio decreases, increasing $\kappa$ values. Organic acids are also being formed and can contribute to increased $\kappa$ values (see further discussion in section 3.4.2). According to (Mikhailov et al., 2013), $\kappa$ is inversely proportional to molar mass, and at equal particle density and at unity van't Hoff factor, a decrease in molar mass would lead to an increase in $\kappa$."

Page 13, line 25: What are the units referring to carbon loss? Is it by mass, volume, molar concentration?

It is by carbon molar concentration. To clarify, we have modified the sentence and it now reads, "The organic carbon loss was dependent on the DOM sample and ranged between 15 % (SRFA) and 63 % (DS 2014) over the course of 25 h of UVB exposure, representing 230 to 740 $\mu$M of carbon loss (Figure 4)."

It's difficult for me to make a clear connection between the increase in hygroscopicity and the production of small organic acids/loss of carbon, except that the author's state there is a dependency of CCN activity on the decrease in organic carbon content. Is the reasoning that the production of small organic acids increases average O/C of the particles and thus κ? Is there any evidence that shows such an enhancement in CCN activity attributed to an increase in O/C and how much of a change in O/C is expected? Perhaps the significant loss of carbon (up to 63%) during irradiation, likely due to volatilization, decreases average DOM molar mass, thus increasing κ, e.g., as in Slade et al. (2017) for Suwannee River Fulvic Acid particles reacting with OH? The following relationship relates molar mass to κ (Mikhailov et al., 2013):

κ=J*(density/M)/(density_w/M_w )

Where J is the van't Hoff factor, M is average particle molar mass, and the subscript "w" refers to water. Assuming constant particle density and unity van't Hoff factor, a decrease in molar mass would increase hygroscopicity. I understand the authors did not analyze the mixtures using mass spectrometry, however, performing such a calculation relating κ to molar mass could help support the argument even further.

We thank the reviewer for this additional insight. We think that we discuss the chemical changes further in section 3.4.5., but that the CCN discussion on 3.2 could have been made clearer. To that effect, we've added the following discussion and reference to section 3.2. It now reads, " We further note the seemingly large $\kappa$ values for organic matter, specifically for the irradiated field collected samples DS 2014, DS 2016 and SR 2017 (Figure 1). We suggest that as the organic matter is photomineralized, the organic-to-inorganic ratio decreases, increasing $\kappa$ values. Organic acids are also being formed and can contribute to increased $\kappa$ values (see further discussion in section 3.4.2). According to (Mikhailov et al., 2013), $\kappa$ is inversely proportional to molar mass, and at equal particle density and at unity van't Hoff factor, a decrease in molar mass would lead to an increase in $\kappa$."

Page 13, line 21: I found the result that pH of the solution remained at ~5 during the irradiation quite surprising. If more organic acids are generated with increasing irradiation, would not the pH further decrease?

Admittedly, we were also surprised. Nonetheless, we hypothesize that DOM has a high enough buffering capacity to minimize pH changes due to weak acids (with pK$_a$s between 2 and 5) during photomineralization. To address the reviewer's point, we have added the following sentence to the text, "We further hypothesize that DOM has a high enough buffering capacity to prevent weak acids from decreasing the pH during photomineralization."

The paper could really benefit from an overall schematic of the proposed processes (mechanisms) occurring. For example, a reaction scheme that shows the production of oxidants internal of the solution, the oxidation of a representative DOM structure and formation of the small organic acids, etc. There are many (exciting and cross-cutting) points discussed in the paper. However, connecting them without a clear schematic, makes it very difficult to follow in the paper.

We thank the reviewer for their comment and suggestion. We further agree that a scheme would be helpful and have now included Figure 7. However, we opt not to display a representative DOM chemical structure. We know that DOM is a combination of lignin, protein, fatty acids, ions, and frankly complex organic matter and we do not want to inadvertently brand DOM as being one specific structure. We made Figure 7 to summarize in a quantitative manner the results presented in this work.

References

Lignell, H., Hinks, M. L., Nizkorodov, S. A.: Exploring matrix effects on photo- chemistry of organic aerosols, Proc. Natl. Acad. Sci. USA, 111, 13780-13785, 10.1073/pnas.1322106111, 2014.

[revised manuscript text omitted]